# Deep Neural Networks for Dental Implant System Classification

**DOI:** 10.3390/biom10070984

**Published:** 2020-07-01

**Authors:** Shintaro Sukegawa, Kazumasa Yoshii, Takeshi Hara, Katsusuke Yamashita, Keisuke Nakano, Norio Yamamoto, Hitoshi Nagatsuka, Yoshihiko Furuki

**Affiliations:** 1Department of Oral and Maxillofacial Surgery, Kagawa Prefectural Central Hospital, 1-2-1, Asahi-machi, Takamatsu, Kagawa 760-8557, Japan; furukiy@ma.pikara.ne.jp; 2Department of Oral Pathology and Medicine, Okayama University Graduate School of Medicine, Dentistry and Pharmaceutical Sciences, Okayama 700-8558, Japan; pir19btp@okayama-u.ac.jp (K.N.); jin@okayama-u.ac.jp (H.N.); 3Department of Electrical, Electronic and Computer Engineering, Faculty of Engineering, Gifu University, 1-1 Yanagido, Gifu, Gifu 501-1193, Japan; yoshii@fjt.info.gifu-u.ac.jp (K.Y.); takeshi.hara@mac.com (T.H.); 4Polytechnic Center Kagawa, 2-4-3, Hananomiya-cho, Takamatsu, Kagawa 761-8063, Japan; kazamakura.ka2suke@gmail.com; 5Department of Orthopaedic Surgery, Kagawa Prefectural Central Hospital, Takamatsu, Kagawa 760-8557, Japan; lovescaffe@yahoo.co.jp

**Keywords:** dental implant, artificial intelligence, classification, deep learning, convolutional neural networks

## Abstract

In this study, we used panoramic X-ray images to classify and clarify the accuracy of different dental implant brands via deep convolutional neural networks (CNNs) with transfer-learning strategies. For objective labeling, 8859 implant images of 11 implant systems were used from digital panoramic radiographs obtained from patients who underwent dental implant treatment at Kagawa Prefectural Central Hospital, Japan, between 2005 and 2019. Five deep CNN models (specifically, a basic CNN with three convolutional layers, VGG16 and VGG19 transfer-learning models, and finely tuned VGG16 and VGG19) were evaluated for implant classification. Among the five models, the finely tuned VGG16 model exhibited the highest implant classification performance. The finely tuned VGG19 was second best, followed by the normal transfer-learning VGG16. We confirmed that the finely tuned VGG16 and VGG19 CNNs could accurately classify dental implant systems from 11 types of panoramic X-ray images.

## 1. Introduction

Osseointegration involves the direct structural and functional connection between living bone and the surface of a load-bearing artificial implant. In dentistry, such implants provide promising prosthetic restoration alternatives [1]. The capability to provide dental implants has revolutionized dental practices worldwide, improving the lives of many patients. The widespread use of dental implants is being improved by technological innovations that mitigate long-term prognoses and the risk of poor alveolar bone conditions. Such innovations include the development of new implant surface textures [2,3] and shapes (e.g., threading [4,5] and platforms [6]) that further improve alveolar ridge augmentation and sinus lift surgeries as pre-implant procedures for alveolar bone atrophy cases [7,8,9]. The growing demand for dental implants has led many manufacturers to enter the industry. Since the year 2000, more than 220 implant brands have been available in the worldwide market [10], and the variety continues to grow.

Implants consist of fixtures, abutments, and superstructures that can vary in terms of style, structure, and tools required, rendering the classification of implant brands difficult. For example, a manufacturers’ proprietary prosthesis fixing screws directly influence implant maintenance (e.g., retightening to counter loosening) [11]. Thus, accurate identification of the implant brand is important. The types of dental implants used and their screws change over time, and different types of implants are often placed by different dentists for a single patient. The difficulty of identification is compounded when information needs to be shared across different countries or regions. With panoramic radiography, it is possible to obtain information related to the jawbone and teeth in one image [12]. Such images often provide the information needed to identify a patient’s implant brand(s). However, doing so requires a significant amount of human effort and experience. No automated method has yet been proposed for identifying implant brands from panoramic radiograph images [13]. Clinically, this capability would be useful.

Artificial intelligence (AI) requires the use of intelligent, machine-based algorithms that mimic human neurological processes. In recent decades, AI has made significant progress in enabling machines to automatically process and categorize complex data [14]. In particular, convolutional neural networks (CNNs), the latest core model of artificial neural networks and deep learning, provide computer-vision capabilities [15], including medical image classification. CNN-based computer-vision technology has produced impressive diagnostic and predictive results in radiology and pathology research and has potential for meeting dental implant-recognition needs [16,17]. AI and deep learning techniques have already been used to support dentistry [18,19].

However, collecting a large amount of image data from clinics can be difficult, and not providing sufficient data for CNNs can lead to overfitting. For these reasons, transfer learning and fine-tuning techniques have been used in recent years. In transfer learning, an existing learned model is used as a feature extractor without changing the weight data, while in fine tuning, an existing learned model is used as a feature extractor by relearning some of the weight data. These methods are powerful methods for training deep CNNs without overfitting, even when the target dataset is smaller than the base dataset [20].

In this study, we assessed the accuracy of using digital panoramic X-ray radiograph images for classification and clarification of dental implant brands via deep convolutional network transfer-learning and fine-tuning strategies.

## 2. Patients and Methods

### 2.1. Study Design

We leveraged a dataset of segmented panoramic radiographs. Several different CNNs were used to classify dental implant brands based on patient panoramic radiographs and ground truth data. We then examined the classification accuracy of a variety of CNN models based on the metric goals set forth in the Introduction.

### 2.2. Performance Metrics

Our primary performance variable was classification accuracy, which corresponds to the proportion of correct classifications. Secondary metrics included the visualization of CNN-focused features into related image regions. The accuracy, precision, recall, recover operating characteristic (ROC) curve, and *F*_1_ score, which consider the relationship between the data’s positive dental implant labels and those given by a classifier, are calculated as follows (Equations (1)–(4)) with the testing dataset using a confusion matrix:(1)$$\mathrm{Accuracy}\;=\frac{\mathrm{TP}\;+\;\mathrm{TN}}{\mathrm{TP}\;+\;\mathrm{FP}\;+\;\mathrm{FN}\;+\;\mathrm{TN}}$$
(2)$$\mathrm{Precision}\;=\frac{\mathrm{TP}}{\mathrm{TP}\;+\;\mathrm{FP}}$$
(3)$$\mathrm{Recall}\;=\frac{\mathrm{TP}}{\mathrm{TP}\;+\;\mathrm{FN}}$$
(4)$$F1\;\mathrm{Score}\;=\frac{2\times \left(\mathrm{Recall}\;\times \;\mathrm{Precision}\right)}{\mathrm{Recall}\;+\;\mathrm{Precision}}$$
where *TP* is true positive, *FP* is false positive, *FN* is false negative, and *TN* is true negative. The area under the ROC curve (AUC) was also calculated.

### 2.3. Ethics Statement

This study was approved by the institutional review board (IRB) of Kagawa Prefectural Central Hospital (Approval No. 849). The IRB waived the need for individual informed consent. Thus, written/verbal informed consent was not obtained from any participant because this study featured a non-interventional retrospective design, and all data were analyzed anonymously. 

### 2.4. Data Preprocessing

Anonymized dental implant radiographic image datasets, acquired between January 2005 and December 2019, were obtained from the picture archiving and communication system of Kagawa Prefectural Central Hospital (HOPE Dr ABLE-GX, FUJITSU Co., Tokyo, Japan) and classified and labeled based on electronic medical records and the dental implant usage ledger of our department. Digital panoramic dental radiographs collected using AZ3000CMR (ASAHIROENTGEN IND. Co., Ltd., Kyoto, Japan) were exported as portable network graphics (PNG) images. From a collection of 6513 selected digital panoramic dental radiographs, a dataset of 8859 manually cropped image segments, each focused on a dental implant, was synthesized. Each included dental implant image was cropped for each dental panoramic radiograph taken as needed. The 11 systems mainly used at Kagawa Prefectural Central Hospital were selected as the dental implants targeted in this study. The types of dental implant systems and corresponding number of images are shown in Table 1. The size of each panoramic X-ray photograph was 2964 × 1464 pixels. Among them, images containing the following 11 types of dental implant systems were selected for this work:Full OSSEOTITE 4.0: Full OSSEOTITE Tapered Certain (Zimmer Biomet, Florida, USA), diameter of 4 mm; lengths of 8.5, 10, 11, and 11.5 mm.Astra EV 4.2: Astra Tech Implant System OsseoSpeed EV (Dentsply IH AB, Molndal, Sweden), diameter of 4.2 mm; lengths of 9 and 11 mm.Astra TX 4.0: Astra Tech Implant System OsseoSpeed TX (Dentsply IH AB, Molndal, Sweden), diameter of 4 mm; lengths of 8, 9, and 11 mm.Astra TX 4.5: Astra Tech Implant System OsseoSpeed TX (Dentsply IH AB, Molndal, Sweden), diameter of 4.5 mm; lengths of 9 and 11 mm.Astra MicroThread 4.0: Astra Tech Implant System MicroThread, (Dentsply IH AB, Molndal, Sweden), diameter of 4 mm; lengths of 8, 9, and 11 mm.Astra MicroThread 4.5: Astra Tech Implant System MicroThread, (Dentsply IH AB, Molndal, Sweden), diameter of 4.5 mm; lengths of 9 and 11 mm.Brånemark Mk III 4.0: Brånemark System Mk III TiUnite (Nobelbiocare, Göteborg, Sweden), diameter of 4 mm; lengths of 8.5, 10, and 11.5 mm.FINESIA 4.2: FINESIA BL HA TP (KYOCERA Co., Kyoto, Japan), diameter of 4.2 mm; lengths of 8 and 10 mmReplace Select Tapered 4.3: Replace Select Tapered (Nobelbiocare, Göteborg, Sweden), diameter of 4.3 mm; lengths of 8, 10, and 11.5 mm.Nobel Replace CC 4.3: NobelReplace Conical Connection, (Nobelbiocare, Göteborg, Sweden), diameter of 4.3 mm; lengths of 8, 10, and 11.5 mm.Straumann Tissue 4.1: Standard Plus Implant Tissue Level implants (Straumann Group, Basei, Switzerland), diameter of 4.1 mm; lengths of 8 and 10 mm.

These dental implant data included implant fixtures, healing abutments, provisional settings, and final prostheses. As preparation before analysis, we used Photoshop Element (Adobe Systems, Inc., San Jose, CA, USA) to manipulate the images so that all dental implant fixtures would fit (see Figure 1 and Figure 2).

### 2.5. Convolutional Neural Network

We used three CNN structures: a basic CNN with three convolution layers, VGG16, and VGG19. Transfer learning and fine tuning were each performed for VGG16 and VGG19. VGG16, which was introduced by the Visual Geometry Group at Oxford University, provides a weighted-layer depth consisting of 13 and 3 fully connected layers [21]. This network was trained on over 1 million images in 1000 classes with more than 370,000 iterations to calibrate 138 million weight parameters. Notably, VGG19 won first place in a classification and localization competition at the Large Scale Visual Recognition Challenge in 2014, a global image-recognition competition. VGG19 has 19 layers with weights. In this model, 16 of 19 layers are convolutional and are divided into five blocks by the max pooling layer [21]. The basic CNN and transfer-learning VGG16/VGG19 models were set to 0.001 and fine-tuning VGG16/19 to 0.0001 for learning rates.

In total, we employed five CNN study groups as follows (Figure 3):Basic CNN model with six convolutional layers (basic CNN)Transfer-learning VGG16 model with pre-trained weights (VGG16 transfer)Transfer-learning and fine-tuning VGG16 model with pre-trained weights (VGG16 fine tuning)Transfer-learning VGG19 model with pre-trained weights (VGG19 transfer)Transfer learning and fine-tuning VGG19 model with pre-trained weights (VGG19 fine tuning)

The datasets were split at the patient image level into 75% training and 25% testing for the different stages of learning performed in this study. The optimization algorithm used for the basic CNN was Adam and for the four VGG models, Momentum SGD. The datasets were trained using transfer learning. The training dataset was separated randomly into 128 batches for every epoch, the number of iterations (epochs) was set at a maximum of 700 from the behavior of validation loss. To evaluate the performance of the current method, fourfold cross-validation was used. As a method to prevent overfitting, generalization is guaranteed by this cross-validation method. This process was repeated for each architecture (i.e., basic CNN, VGG16/VGG19 transfer, and VGG16/VGG19 fine tuning). All models were trained and evaluated on a 64-bit Ubuntu 16.04.5 LTS operating system with 31.4 GB memory and an NVIDIA GeForce GTX TITAN X graphics processing unit. Building, training, and prediction of deep-learning models were performed using the Keras library (https://keras.io) and TensorFlow [22] back-end engine.

### 2.6. Model Visualization

CNN model visualization helps to clarify the most relevant features used for classification. To identify potential correct classifications based on incorrect features and to gain some intuition into the classification process, we identified the image pixels most relevant for classification using gradient-weighted class activation maps (Grad-CAM) [23]. Map visualizations are heatmaps of the gradients with the “hotter” colors representing the regions of more importance for classification. The heat map using Grad-CAM was reconstructed with the final convolutional layer in this study.

## 3. Results

### 3.1. Classification Performance

The CNN models used in this study were trained using the cross-entropy loss function of the selected training image dataset. The image classification performance each of the five CNN models tested in this study is shown in Table 2.

The finely tuned VGG16 model with pretrained weights achieved the best performance for all metrics, including recall, precision, accuracy, and F-measure. The next best performer was the finely tuned VGG19 model. There were no clear differences between the finely tuned VGG16 and the finely tuned VGG19, but the performance of the finely tuned VGG16 was slightly better. The next best performance was the VGG16 with transfer learning, followed by the VGG19 with transfer learning and the basic CNN. The F_1_ score performance per CNN model for each dental implant classification is shown in Table 3.

The finely tuned VGG16 performed best in 9 of 11 dental implant categories. However, the finely tuned VGG19 performed better overall. Straumann Tissue 4.1 implants performed well on all models, and Astra TX 4.5 and Astra MicroThread 4.5 implants performed slightly lower overall than the others.

Table 4 shows the dental implant classification performance of all tested models by AUC. Fine tuning of VGG16 and VGG19 showed a high AUC, with the basic CNN having the lowest AUC for all dental implants. The ROC curve is shown in Appendix A.

### 3.2. Visualization of Model Classification

Figure 4 shows images of the 11 types of dental implants classified using each CNN model, visualized using Grad-CAM. The finely tuned VGG16 and finely tuned VGG19 both showed an identification area that could be used to identify similar images. The transfer-learning VGG16 and transfer-learning VGG19 both showed the identification areas used to identify similar images. The basic CNN indicated only the outline of the implant as the identification area. From our results, the discriminative region distinguished the implant system from not only the implant fixture but also the entire circumference. In particular, we observed that both finely tuned CNNs discriminated not only using part of the fixture but also using the whole. Visualization images for each dental implant system are shown in Appendix A.

## 4. Discussion

We demonstrated that the five CNNs surveyed were able to classify 11 dental implant systems extracted from panoramic X-ray images with high accuracy despite mixed conditions during the implant-treatment stage. Grad-CAMs for each network were also found to understand the characteristics of each convolutional layer for each implant fixture. These results played an important role in classifying dental implant brands from panoramic radiographs via deep learning.

By applying appropriate transfer learning and fine-tuning to the pre-trained deep CNN architectures, we were able to perform image classification with high accuracy when using relatively small image datasets. The classification performance of basic CNNs with only three convolutional layers was the worst. We thus inferred that CNN models with few convolutional layers have limited machine-learning capability for approximately 10 image classifications involving a small dataset. The results of this study also showed that fine-tuning some convolutional blocks in the deep CNN layers could improve image classification performance. Generally, deep CNN models trained from pretrained deep neural networks on large natural image datasets are good for general image classification. However, they are not effective for specific classifications, such as those for medical imaging. The findings show that if a particular convolutional block of a deep CNN model is finely tuned, deep CNN can be more specialized for a particular classification task [24]. This is an important finding that shows the usefulness of fine adjustments in medical imaging.

A disadvantage of CNNs is that they are black boxes that cannot explain the characteristics of machine learning and the grounds for making decisions based on that learning [25]. Therefore, feature visualization using Grad-CAM was applied. This process helps humans understand that features or areas of images are used for classification decisions [26]. Our visualization results were also interesting. Eleven types of implants were the subject of this classification. Among them, various states were classified (e.g., only implant fixtures, fixtures with abutments, and fixtures with superstructure). Dentists typically identify fixtures from the conditions of the implants and brands from their morphologies. CNNs, as we have shown, can also be used to perform the same feature extractions. However, we found cases in which some implants were classified by treating the entire background as a feature (see Astra MicroThread 4.0 results). Notably, apart from the different feature extraction processes, there was no difference in the accuracy of classification. The features around the implant can thus be used to characterize the overall morphology of the fixture.

The main advantage of panoramic radiography is the ability to detect tooth- and jaw-related objects simultaneously [27]. Despite the plethora of images available, few studies [19,28,29,30,31] have applied CNNs to their classifications and diagnoses. Studies that used panoramic radiographs often involved diseases related to the jawbone [28,29,31] and the maxillary sinus [19]. Because panoramic radiographs have different distortions depending on the region to be photographed, periapical radiographic images have generally been used for diagnosis, whereas CNNs have been used for tooth-related classifications and diagnoses [32,33]. The same was found to be true for the classification of dental implants using CNNs [13]. From our study, we found that CNNs using panoramas lead to results comparable to the diagnostic accuracy of dental implant classification by CNNs using periapical radiographic images [13]. These results will contribute to the accuracy of CNN classification diagnoses by increasing the number of images used via preprocessing.

Compatibility is different depending on the dental implant system [34]. Some systems are not compatible with other brands, whereas others are broadly compatible. As mentioned, these factors directly affect the maintenance of implant prostheses. Nevertheless, patient implant maintenance will continue as long as the device remains in the patient’s oral cavity. The implant systems examined with this study were developed in 2020. It is important to accumulate current data and to use the learned network for the next generation of devices. Even if it is difficult to obtain information on implant systems that have been discontinued, it is necessary for dentists to prepare those systems to easily procure and respond to implant data that have been accumulated thus far.

We showed that deep neural networks are suitable for classifying the included dental implants. There is potential to apply them to more implant systems [10] and setups with different devices for image acquisition. Major dental implant systems and radiography devices differ in every region of the world. Therefore, it is first necessary to create an accurate database for each and build accurate classification by deep neural network based on the database. We hope that cross-sectional studies by other institutions in the world will help to build a stronger dental implant classification method using CNNs.

This study had three limitations. The first was the narrow selection of CNN models. The commonly used VGG16 and VGG19 models were employed. Deep-learning algorithms (e.g., ResNet and CapsNet) with deep and wide layers or those with modified stratification methods are being continuously developed [35]. It will thus be necessary to study these and various other CNN models in the future. Second, the X-ray images used for classification in this study were taken with the same panoramic X-ray equipment. With different panoramic X-ray equipment, the image quality and magnification provided will vary. Therefore, in future work, it will be necessary to estimate the results in a large cross-sectional study involving various panoramic radiographs and image qualities. Third, we evaluated image segments manually cropped from panoramic radiographs. Creating a network that can detect implants using uncropped panoramic images or those that can apply techniques to detect multiple implants simultaneously would be a more valuable direction for future research.

## 5. Conclusions

In our study, we demonstrated that the deep CNNs surveyed were able to classify 11 dental implant systems extracted from digital panoramic X-ray images with high accuracy, despite mixed conditions at the implant-treatment stage. In particular, VGG16 and VGG19 finely tuned CNNs showed excellent classification performance. Grad-CAM for each network was also shown to understand the characteristics of each convolutional layer for each implant fixture. These results will play an important role in determining dental implant brands from panoramic radiographs using deep learning.

## Figures and Tables

**Figure 1 biomolecules-10-00984-f001:**
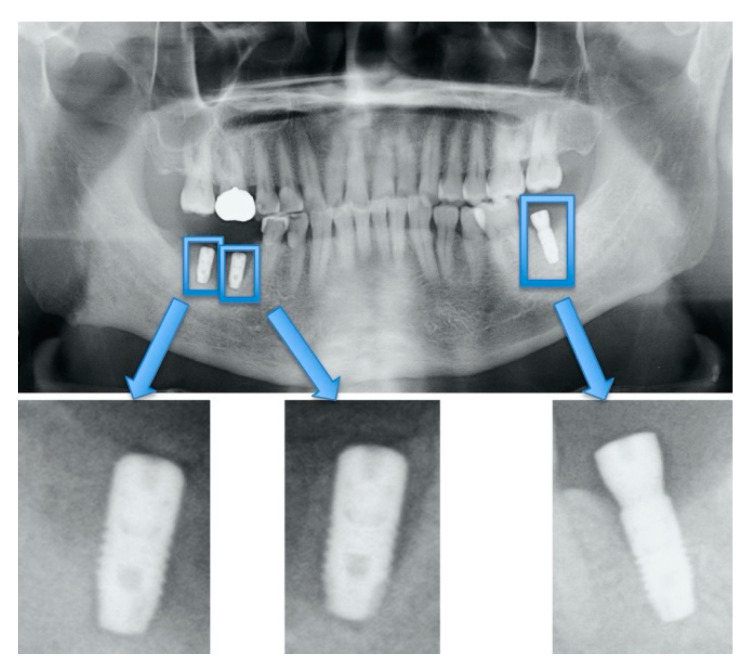
Cropping of dental implant imagery to include single fixtures.

**Figure 2 biomolecules-10-00984-f002:**
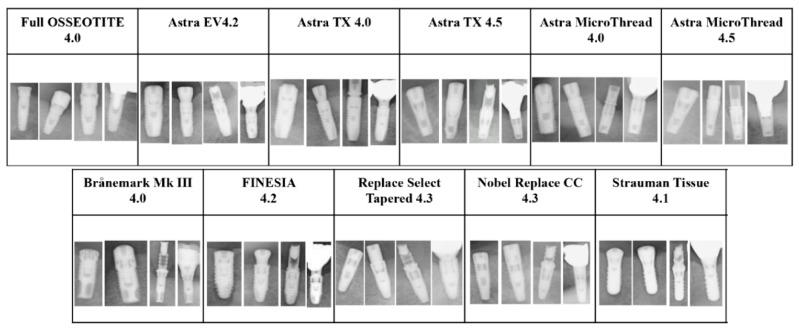
Eleven types of dental implant systems cropped from panoramic radiographs. The images of each system include implant fixtures, dental implants with healing abutments, dental implants with provisional settings, and implants with final prostheses.

**Figure 3 biomolecules-10-00984-f003:**
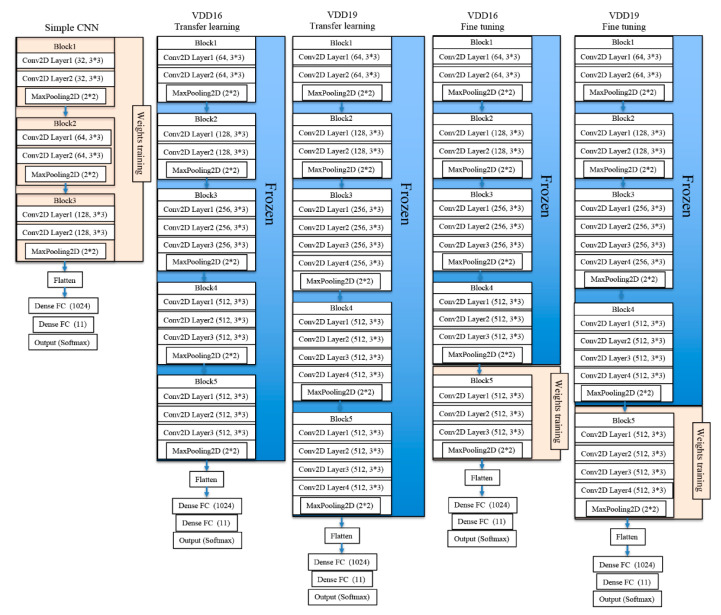
Schematic of the five convolutional neural networks (CNN) architectures.

**Figure 4 biomolecules-10-00984-f004:**
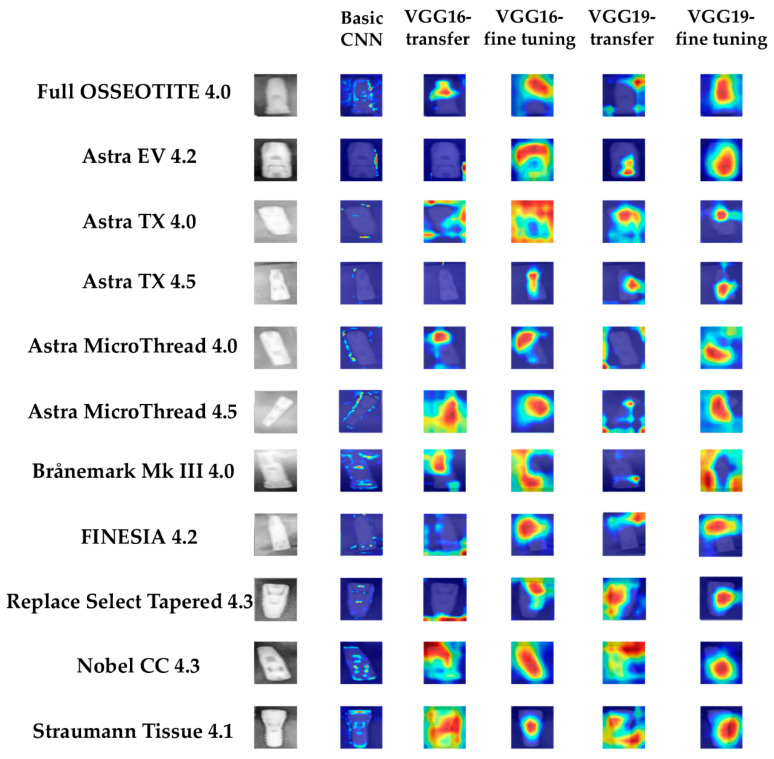
Example of the class activation maps of the five CNN networks for the eleven dental implant systems.

**Table 1 biomolecules-10-00984-t001:** Types of dental implant systems and corresponding number of images.

Dental Implant System	Full OSSEOTITE 4.0	Astra EV 4.2	Astra TX 4.0	Astra MicroThread 4.0	Astra MicroThread 4.5	Astra TX 4.5
Company	Biomet	Dentsply	Dentsply	Dentsply	Dentsply	Dentsply
Diameter (mm)	4.0	4.2	4.0	4.0	4.5	4.5
Length (mm)	8.5	8.0	8.0	8.0	9.0	9.0
	10.0	9.0	9.0	9.0	11.0	11.0
	11.0	11.0	11.0	11.0		
	11.5					
Number of images	427	425	2521	1088	698	387
Implant fixture	278	201	1416	512	332	226
Implants with healing abutment	25	152	506	156	80	94
Prostheses	124	72	599	420	286	67
**Dental implant system**	**Brånemark Mk III 4.0**	**FINESIA 4.2**	**Replace Select Tapered 4.3**	**Nobel CC 4.3**	**Straumann Tissue 4.1**	
Company	Nobelbiocare	KYOCERA	Nobelbiocare	Nobelbiocare	Straumann	
Diameter (mm)	4.0	4.2	4.3	4.3	4.1	
Length (mm)	8.5	8.0	8.0	8.0	8.0	
	10.0	10.0	10.0	10.0	10.0	
	11.5		11.5	11.5		
Number of images	423	233	486	1681	490	
Implant fixture	255	105	202	1073	199	
Implants with healing abutment	146	101	145	155	211	
Prostheses	22	27	139	453	80	

**Table 2 biomolecules-10-00984-t002:** Dental implant classification accuracy of CNN models.

	Recall	Precision	Accuracy	F-measure
**Basic CNN**	0.802	0.842	0.860	0.819
**VGG16-transfer**	0.864	0.888	0.899	0.874
**VGG16-fine tuning**	0.907	0.928	0.935	0.916
**VGG19-transfer**	0.840	0.873	0.880	0.853
**VGG19-fine tuning**	0.894	0.913	0.927	0.902

**Table 3 biomolecules-10-00984-t003:** Dental implant classification performance by F_1_ score.

	Full OSSEOTITE 4.0	Astra EV 4.2	Astra TX 4.5	Astra MicroThread 4.0	Astra MicroThread 4.5	Astra TX 4.0
**Basic CNN**	0.849	0.701	0.658	0.778	0.746	0.930
**VGG16-transfer**	0.899	0.799	0.739	0.879	0.815	0.938
**VGG16-fine tuning**	0.955	0.860	0.770	0.928	0.866	0.969
**VGG19-transfer**	0.874	0.765	0.705	0.837	0.819	0.918
**VGG16-fine tuning**	0.953	0.831	0.740	0.917	0.890	0.961
	**Brånemark** **Mk III 4.0**	**FINESIA 4.2**	**Replace Select Tapered 4.3**	**Nobel CC 4.3**	**Straumann Tissue 4.1**	
**Basic CNN**	0.871	0.831	0.805	0.933	0.905	
**VGG16-transfer**	0.910	0.931	0.801	0.944	0.962	
**VGG16-fine tuning**	0.935	0.966	0.876	0.969	0.986	
**VGG19-transfer**	0.879	0.898	0.797	0.921	0.970	
**VGG16-fine tuning**	0.940	0.915	0.836	0.961	0.983	

**Table 4 biomolecules-10-00984-t004:** Dental implant classification performance by the area under the recover operating characteristic curve (AUC).

	Full OSSEOTITE 4.0	Astra EV 4.2	Astra TX 4.5	Astra MicroThread 4.0	Astra MicroThread 4.5	Astra TX 4.0
**Basic CNN**	0.986	0.959	0.958	0.978	0.969	0.986
**VGG16-transfer**	0.999	0.991	0.987	0.996	0.993	0.998
**VGG16- fine tuning**	0.997	0.979	0.981	0.989	0.987	0.994
**VGG19-transfer**	0.999	0.992	0.987	0.995	0.992	0.998
**VGG16- fine tuning**	0.993	0.975	0.980	0.987	0.984	0.991
	**Brånemark** **Mk III 4.0**	**FINESIA 4.2**	**Replace Select Tapered 4.3**	**Nobel CC 4.3**	**Straumann Tissue 4.1**	
**Basic CNN**	0.988	0.994	0.981	0.993	0.993	
**VGG16-transfer**	0.998	0.999	0.995	0.998	1.000	
**VGG16- fine tuning**	0.996	0.999	0.984	0.996	0.999	
**VGG19-transfer**	0.997	1.000	0.990	0.998	1.000	
**VGG16- fine tuning**	0.997	0.997	0.981	0.993	0.999

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
