# Peer review of "Deep Neural Networks for Dental Implant System Classification"

_biomolecules, 2020, doi:10.3390/biom10070984_

Round 1
Reviewer 1 Report
Dear Authors,
Thank you very much for submitting this interesting manuscript entitled „ Deep Neural Networks for Dental Implant System Classification”. The automatic screening of panoramic radiographs may improve cross-sectional studies in future and may help to get lager sample sizes. Hence, the CNN may represent a paradigm shift. According to your setup and study design, you proved that CNN is suited to identify the included dental implants.
Nevertheless, there are some aspects that should be clarified or better discussed:
1) Dataset collection should be better described. I assume that all radiographs had an adequate quality and were therefore suited to identify dental implants systems.
All the radiographs were preformed using only one device AZ3000CMR, ASAHIROENTGEN IND. Co., Ltd., Kyoto, Japan. This is definitively a limitation of your study. You always had a nearly identical image quality when analyzing the data.
Different devices may offer a different image quality and different magnification factors.
You should mention this limitation – actually, it remains unclear whether the results can be extrapolated for larger cross-sectional studies including different devices and image qualities.
2) What kind of format was used analyzing the images? RAW, TIFF; JPEG – Did you used compressed data? Nothing is mentioned concerning this aspect.
3) Inclusion criteria: 8,859 manually cropped image segments, each focusing on the dental implant, from 6,513 chosen digital panoramic dental radiographs collected. It remains unclear if each implant was only included only once. It could be included multiple times into the sample – taking into account that patient usually get more than one panoramic radiograph when dental implant were inserted (immediately post-operative, before implant exposure, in function with prosthodontic coverage).
4) In the exemplary pictures, most of the implant were without healing abutments or prosthetic restorations. Hence, I assume that mainly images immediately post-operative were included. I would like to have more details concerning the included x-rays.
5) Inclusion criteria: Nothing is mentioned concerning the included dental implants. Is there any reason for these selection? More than 220 different implant systems are available – hence this selection should be justified.
6) Of course, the Deep Neural Networks are suited to classify the included dental implants, but at the current stage, it is not proven that the results may be extrapolated to more implant systems and a setup using different devices for image acquisition.
Hence, to my point of view, it is a feasibility study.
This aspect should be implemented in the title and should be thoroughly discussed.
Apart from that, this manuscript is well prepared and offers future perspectives in digital image analyzing. Future cross-sectional studies may have larger sample sizes and therefore more power.
Author Response
Responses to Reviewers’ Comments
Thank you very much for your invaluable comments and kind acceptance. We have incorporated all the reviewers’ comments and suggestions into our manuscript; the corresponding changes are highlighted in red font in the revised manuscript.
We would like to say thank you once again for the reviewers’ suggestions, which were very helpful in further improving our manuscript.
Comments from Reviewers and Responses
Reviewer 1
Thank you very much for submitting this interesting manuscript entitled Deep Neural Networks for Dental Implant System Classification”. The automatic screening of panoramic radiographs may improve cross-sectional studies in future and may help to get lager sample sizes. Hence, the CNN may represent a paradigm shift. According to your setup and study design, you proved that CNN is suited to identify the included dental implants.
Nevertheless, there are some aspects that should be clarified or better discussed:
Comment 1) Reviewer1: “Dataset collection should be better described. I assume that all radiographs had an adequate quality and were therefore suited to identify dental implants systems.
All the radiographs were preformed using only one device AZ3000CMR, ASAHIROENTGEN IND. Co., Ltd., Kyoto, Japan. This is definitively a limitation of your study. You always had a nearly identical image quality when analyzing the data.
Different devices may offer a different image quality and different magnification factors.
You should mention this limitation – actually, it remains unclear whether the results can be extrapolated for larger cross-sectional studies including different devices and image qualities.”
Response:
We thank you for this helpful comment. The limitation you pointed out is indeed correct; data collection by the same panoramic imager is one research limitation in this study. We have added this limitation in the discussion section.
Comment 2) Reviewer1: What kind of format was used analyzing the images? RAW, TIFF; JPEG – Did you used compressed data? Nothing is mentioned concerning this aspect.
Response:
We thank you for this helpful comment. We converted the dental panorama x-ray data into PNG images. We added this information to the Data Preprocessing section of Patients and Methods.
Comment 3) Reviewer1: Inclusion criteria: 8,859 manually cropped image segments, each focusing on the dental implant, from 6,513 chosen digital panoramic dental radiographs collected. It remains unclear if each implant was only included only once. It could be included multiple times into the sample – taking into account that patient usually get more than one panoramic radiograph when dental implant were inserted (immediately post-operative, before implant exposure, in function with prosthodontic coverage).
Response:
Thank you for this helpful comment. Each included dental implant image was cropped for each dental panoramic radiograph taken as needed. The same implant was cropped for each panoramic radiograph taken in each condition. We have added this information to the Data Preprocessing section of Patients and Methods.
Comment 4) Reviewer1: In the exemplary pictures, most of the implant were without healing abutments or prosthetic restorations. Hence, I assume that mainly images immediately post-operative were included. I would like to have more details concerning the included x-rays.
Response:
Thank you for this helpful comment. It is as you inferred. The details of the implant images have been added to Table 1. The image by Grad-cam is only Fixture in the text, but an image including abutment and prosthesis is included in the Supplementary Materials.
Comment 5) Reviewer1: Inclusion criteria: Nothing is mentioned concerning the included dental implants. Is there any reason for these selection? More than 220 different implant systems are available – hence this selection should be justified.
Response:
Thank you for this helpful suggestion. The eleven systems mainly used at Kagawa Prefectural Central Hospital were selected as the dental implants targeted in this study. We have added this information to the Data Preprocessing section of Patients and Methods.
Comment 6) Reviewer1: Of course, the Deep Neural Networks are suited to classify the included dental implants, but at the current stage, it is not proven that the results may be extrapolated to more implant systems and a setup using different devices for image acquisition.
Response:
Thank you for this helpful comment. It is as you pointed out. Construction of CNN for classification of dental implants for various implant systems and radiographies is a future prospect. We have added this point.
Reviewer 2 Report
In this manuscript, the authors have utilized five CNN models with panoramic X-ray images to classify 11 types of dental implant system. The main contribution of this study is using deep learning classifiers to analyze X-ray radiograph images for a dental application. The authors clearly outline the clinical context and the problem that they have addressed. However, before the manuscript could be accepted, I have a few concerns and wish the authors could address:
- The authors should briefly describe the CNN models in terms of hyperparameters, learning rates, numbers/sizes of filters, activation functions, number of neurons in each FC layer, etc.
- There is no discussion on how to prevent overfitting using different methods (there are several methods to prevent overfitting such as data augmentation, regularization, etc.). They simply said “CNNs with less data can lead to overfitting. For this reason, transfer learning and fine-tuning techniques have been used in recent years…” “These methods have been shown to be powerful methods for training deep CNNs without overfitting...”
- The authors said “The finely tuned VGG16 model having pretrained weights achieved the best performance” but they do not elucidate how they have reached this conclusion. Based on what numbers? I would suggest performing a proper comparison between CNN models in terms of accuracy, recall, precision, and F-measure in the text, and mention the best accuracy value.
- There are no discussions whatsoever on optimization methods utilized in the backpropagation.
- It would be helpful if the authors expand section 2.6. (Model Visualization) and discuss Grad-CAM in more detail. At what layer(s) these heat maps were reconstructed?
- There are two minor issues. (1) It is preferred to start the introduction with the research motivation not a methodology statement. They may move this sentence “This study assesses the accuracy of using of digital panoramic X-ray radiograph images for…” to the last paragraph of the introduction. (2) Both Eqs. 2 and 3 have the same name “precision.” Please fix that.
Author Response
Responses to Reviewers’ Comments
Thank you very much for your invaluable comments and kind acceptance. We have incorporated all the reviewers’ comments and suggestions into our manuscript; the corresponding changes are highlighted in red font in the revised manuscript.
We would like to say thank you once again for the reviewers’ suggestions, which were very helpful in further improving our manuscript.
Comments from Reviewers and Responses
Reviewer 2
In this manuscript, the authors have utilized five CNN models with panoramic X-ray images to classify 11 types of dental implant system. The main contribution of this study is using deep learning classifiers to analyze X-ray radiograph images for a dental application. The authors clearly outline the clinical context and the problem that they have addressed. However, before the manuscript could be accepted, I have a few concerns and wish the authors could address:
Comment 1) Reviewer2: The authors should briefly describe the CNN models in terms of hyperparameters, learning rates, numbers/sizes of filters, activation functions, number of neurons in each FC layer, etc.
Author response: We thank the reviewer for this helpful comment. We added the learning rates by model. For numbers/sizes of filters, activation functions, number of neurons in each FC layer, we modified Figure 3.
Comment 2) Reviewer2: There is no discussion on how to prevent overfitting using different methods (there are several methods to prevent overfitting such as data augmentation, regularization, etc.). They simply said “CNNs with less data can lead to overfitting. For this reason, transfer learning and fine-tuning techniques have been used in recent years…” “These methods have been shown to be powerful methods for training deep CNNs without overfitting...”
Author response: We thank the reviewer for this helpful comment.
As a method to prevent over-fitting, generalization is guaranteed by four-fold cross-validation. We've added this explanation to Patients and Methods.
Comment 3) Reviewer2: The authors said “The finely tuned VGG16 model having pretrained weights achieved the best performance” but they do not elucidate how they have reached this conclusion. Based on what numbers? I would suggest performing a proper comparison between CNN models in terms of accuracy, recall, precision, and F-measure in the text, and mention the best accuracy value.
Author response: We thank the reviewer for this helpful comment. The finely tuned VGG16 model having pretrained weights achieved the best performance for all metrics including recall, precision, accuracy, and F-measure.
Comment 4) Reviewer2: There are no discussions whatsoever on optimization methods utilized in the backpropagation.
Author response: We thank the reviewer for this helpful comment. The optimization algorithm is Adam for basic CNN and Momentum SGD for the other four VGG models. We've added this explanation to Patients and Methods.
Comment 5) Reviewer2: It would be helpful if the authors expand section 2.6. (Model Visualization) and discuss Grad-CAM in more detail. At what layer(s) these heat maps were reconstructed?
Author response: We thank the reviewer for this helpful comment. The heat map using Grad-CAM was reconstructed with the final convolutional layer. We've added this explanation to Patients and Methods.
Comment 6) Reviewer2: There are two minor issues. (1) It is preferred to start the introduction with the research motivation not a methodology statement. They may move this sentence “This study assesses the accuracy of using of digital panoramic X-ray radiograph images for…” to the last paragraph of the introduction. (2) Both Eqs. 2 and 3 have the same name “precision.” Please fix that.
Author response: We thank the reviewer for this helpful comment.
(1) It is as you pointed out. We moved “This study assesses the accuracy of using of digital panoramic X-ray radiograph images for…” to the end of the introduction.
(2) We have fixed the "precision".
Reviewer 3 Report
The paper presents an arhitecture of deep convolutional networks in an effort to asses the accuracy of classificaion of digital panoramic X-ray radiography images. After an introductory state-of-the-art, the paper describes the experimental setup and the system architecture. The detailed results of five different CNN architectures are then assessed and compared. A discussion of the results, put in balance with the state-of-the-art, follows. The writing style is good, and the use of English language is good. I recommend authors to rewrite the section on authors contribution at page 11. The first two lines should be ommitted. My recommendation is to accept the paper for publication.
Author Response
Responses to Reviewers’ Comments
Thank you very much for your invaluable comments and kind acceptance. We have incorporated all the reviewers’ comments and suggestions into our manuscript; the corresponding changes are highlighted in red font in the revised manuscript.
We would like to say thank you once again for the reviewers’ suggestions, which were very helpful in further improving our manuscript.
Reviewer 3
The paper presents an arhitecture of deep convolutional networks in an effort to asses the accuracy of classificaion of digital panoramic X-ray radiography images. After an introductory state-of-the-art, the paper describes the experimental setup and the system architecture. The detailed results of five different CNN architectures are then assessed and compared. A discussion of the results, put in balance with the state-of-the-art, follows. The writing style is good, and the use of English language is good. I recommend authors to rewrite the section on authors contribution at page 11. The first two lines should be ommitted. My recommendation is to accept the paper for publication.
Response:
We sincerely thank you for the time taken to review our paper and your assessment of this study as excellent. We have deleted the first two lines of the section on author contributions on page 11.
Round 2
Reviewer 1 Report
Dear authors,
Thank you for resubmitting this manuscript and answering all comments point by point and implementing all the demanded revisions. It is always crucial to discuss the limitations of such studies.
The manuscript has improved and is now suited being published.